# Psychological Factors and Sexual Risk Behaviors: A Multidimensional Model Based on the Chilean Population

**DOI:** 10.3390/ijerph19159293

**Published:** 2022-07-29

**Authors:** Rodrigo Ferrer-Urbina, Patricio Mena-Chamorro, Marcos Halty, Geraldy Sepúlveda-Páez

**Affiliations:** Escuela de Psicología y Filosofía, Universidad de Tarapacá, Arica 1020000, Chile; rferrer@academicos.uta.cl (R.F.-U.); pmena@uta.cl (P.M.-C.); mcarmonah@academicos.uta.cl (M.H.)

**Keywords:** human immunodeficiency virus (HIV), sexual risk behaviors, psychological factors

## Abstract

Human immunodeficiency virus (HIV) is a global health problem, with sexual risk behaviors (SRB) being the main routes of spreading the virus. Evidence indicates that different psychological factors influence SRB (e.g., attitude towards condoms, sexual self-concept, sexual sensation seeking, knowledge of sexual risk behaviors, risk perception). This study proposes an explanatory model of sexual risk behaviors in young people and adults. The sample consisted of 992 young people and adults aged between 18 and 35 years. The model presented good levels of fit (X^2^ = 3311.433, df = 1471, CFI = 0.964, TLI = 0.959, RMSEA = 0.036), explaining 56% of the variance of sexual activity with multiple partners, 77% of the inadequate use of protective barriers, and 58.8% of sexual activity under the influence of alcohol or drugs from a set of psychological factors in self-report measures. The details of the results offer novel contextual evidence for the prioritization of prevention-oriented psychosocial interventions.

## 1. Introduction

Human immunodeficiency virus (hereafter, HIV) is a major public health problem, and even though life expectancy has increased following advances in treatments (i.e., antiretroviral therapy) [1], it remains particularly acute in low-income countries [2]. Although the spread of HIV infections has slowed in recent decades, the number of new infections remains alarming, reaching 1.5 million new infections in 2020, concentrated mostly in young people and adults [3,4].

This age focus is attributed to economic, social, and individual factors [5] but mainly to developmental stages where it is more common to have a higher number of sexual partners and to engage in other risky sexual behaviors [6].

Sexual risk behaviors (hereafter, SRB) refer to behaviors that increase the probabilities of the unwanted consequences of sexual activity (e.g., unintended pregnancy, sexually transmitted diseases) [7]. They are the main route of HIV transmission (e.g., [8,9,10]); therefore, to avoid new cases of HIV, it is necessary to prevent high-risk sexual behaviors.

The identified risk behaviors that have evidenced a significant relationship in the literature are the inadequate use of protective barriers, sexual encounters under the influence of alcohol and drugs, and multiple sexual partners [10,11].

The literature has shown different explanatory models of SRBs, such as the planned action theory [12], the health belief model [13], and the protective motivation theory [14]. In particular, these models highlight the relevance of self-efficacy, sexual assertiveness, knowledge, attitudes, and perception of risk and severity, among other variables [15], as associated factors with SRBs [16]. Evidence shows that different safe (e.g., increased condom use) or risky behaviors (e.g., multiple sexual partners; inappropriate condom use) are associated with different psychological factors, including behavioral dispositions or personality traits [17]; ability to identify risky and safe behaviors [18]; perceived vulnerability or risk [19]; sexual self-image [20]; and attitude toward condoms [21]. Specifically, for this study, the following psychological factors were selected.

Attitude toward condoms: an individual’s favorable or unfavorable evaluation of condom use [22]. Attitude toward condom use has frequently been related to SRB [16]; specifically, negative attitudes towards condom use would be an obstacle to the adequate use of protective barriers [18,23], while positive attitudes would increase the likelihood of condom use [24].

Sexual self-concept is people’s thoughts and beliefs that about themselves in the sexual domain [25]. These self-evaluations are associated with sexual risk behaviors [20], showing that when the self-concept is low, there is greater sexual risk-taking, a higher number of sexual partners, and a lower use of condoms [26]. In this direction, it has been suggested that those with a high (higher) self-concept have a greater cognition (awareness) of sexual risk, which translates into a higher number of safe actions [27].

Sexual sensation seeking is a personality trait characterized by a preference for seeking novelty and sexual thrill-seeking experiences to achieve optimal sexual arousal [28]. Evidence suggests that people with higher levels of sexual sensation seeking tend to have a higher number of sexual partners and more permissive attitudes toward sexual encounters [29].

Knowledge of Risky/No-Risky Sexual Behaviors is the degree of information one has about risky behaviors and situations [30]. Evidence suggests that people with a lower ability to identify SRBs are more likely to have sex under the influence of alcohol and without protective barriers [31].

HIV risk perception is the self-perceived likelihood of contracting HIV [32]. Evidence suggests that those who perceive themselves to have a lower HIV risk tend to engage in condomless sex and are likely to have less HIV testing [33].

Given the prevalence that HIV has in the population and the severe consequences it can generate on people’s health [34], multiple research efforts have been conducted to establish effective strategies to prevent SRBs (e.g., [35,36]). Although the variables incorporated in the present study (i.e., attitude towards condoms, sexual self-concept, sexual sensation seeking, knowledge of risky sexual behaviors, risk perception) possess plenty of evidence of their relationship with SRBs, in diverse populations, available studies have been limited to estimate relationships restricted to a smaller number of variables, which prevents an adequate assessment of the joint effects (e.g., [17,37]), especially considering that the independent variables involved can have significant covariation effects (e.g., [38]). This restriction can lead to an overestimation of the effects of the variables studied and, therefore, does not allow adequate prioritization for prevention and intervention programs, with the risk that some of the observed relationships are overestimated or are spurious effects.

Therefore, the current study proposes an explanatory model (see Figure 1) of sexual risk behaviors in young people and adults by analyzing the combined effects of psychological factors (i.e., attitude towards condoms, sexual self-concept, sexual sensation seeking, knowledge of sexual risk behaviors, risk perception), integrating variables from different explanatory models of social and health psychology, on sexual risk behaviors (i.e., inadequate use of protective barriers, sexual activity under the influence of alcohol and drugs, and multiple sexual partners).

## 2. Materials and Methods

### 2.1. Design and Participants

Non-experimental cross-sectional study with a correlational scope [39]. Participants were recruited through a non-probabilistic sampling strategy, by quotas [40], considering the main demographic characteristics (city, age, sex, and educational level) according to the reference proportions granted by the results of the CENSO 2017 [41]. The inclusion criteria were to reside in the study cities and to be of legal age (18 years and older for Chile). All individuals who responded to less than 80% of the questionnaire or presented aberrant response patterns (i.e., surveys without variability, where participants selected the same response option in all items) were excluded from the study.

The valid sample was composed of 992 young people and adults between 18 and 35 years of age, 52.4% (*n* = 514) were women and 47.3% (*n* = 464) were men, from the five main cities of Norte Grande de Chile: Arica (22.0%; *n* = 218), Iquique (14.3%; *n* = 142), Alto Hospicio (9.5%; *n* = 94), Antofagasta (37.1%; *n* = 368), and Calama (17.1%; *n* = 170). Of the total sample, 82.4% (*n* = 818) identified themselves as heterosexual, 45.0% (*n* = 447) reported having been tested for HIV/AIDS, and 37.9% (*n* = 377) reported not having used protective barriers during the last two years. Demographic details are shown in Table 1.

### 2.2. Instruments

*Sexual risk behavior scale* Haga clic o pulse aquí para escribir texto.: 12-item scale, designed to measure four dimensions of sexual risk behaviors: sexual activity with multiple partners (4 items), inappropriate use of protective barriers (4 items), sexual activity under the influence of alcohol or drugs (4 items), and knowledge of the partner’s sexual record (items = 4). Response options corresponded to behavior/attitude statements in a Likert format of four ordered categories (i.e., 0 = “never” to 3 = “always”), which were conditioned to only report behaviors in the past two years. Higher scores suggest a higher frequency of risky sexual behaviors. The scale showed evidence of validity based on the internal test-retest structure and adequate levels of reliability [42].

*Scale of knowledge about HIV risk situations and behaviors* Haga clic o pulse aquí para escribir texto.: This 16-item scale measured two dimensions: knowledge about risky behaviors (6 items) and knowledge about non-risky behaviors (10 items). The knowledge scale was composed of behavioral/attitudinal statements. Some referred to sexual behaviors that constitute real transmission risks, and others referred to interactions with people with HIV/AIDS that do not constitute a risk of transmission. The scale scoring constituted a test of optimal performance, assigning a hit (1) when the rating was adequate and a miss (0) when the rating was inadequate. The scale presented evidence of validity based on the test’s internal structure and adequate levels of reliability [43].

*Condom Use Attitudes Scale* Haga clic o pulse aquí para escribir texto.: 10-item scale which measured the subjective valence of prevention behaviors and the use of protective barriers through three attitudinal dimensions: affective (3 items), behavioral (3 items), and cognitive (4 items). The response options were in a four-category ordered Likert format (i.e., 1 = “Strongly disagree” to 4 = “Strongly agree”). The statements referred to negative attitudes/behaviors toward condom use; therefore, high scores suggest an unfavorable attitude toward condom use. The scale presented evidence of validity based on an internal test–retest structure and adequate levels of reliability [44].

*HIV risk perception scale* Haga clic o pulse aquí para escribir texto.: 9-item scale designed to measure young adults’ perceived HIV risk through two dimensions: perceived HIV susceptibility (4 items) and perceived HIV severity (5 items). Response options corresponded to behavioral/attitudinal statements in a four-ordered Likert format with differential references for perceived HIV susceptibility (i.e., 0 = “false” to 3 = “true”) and perceived HIV severity (i.e., 0 = “not at all” to 3 = “severely”). Higher scores suggest higher levels of perceived HIV risk. The scale presented evidence of validity based on the internal test structure and adequate levels of reliability [45].

*Multidimensional scale of sexual self-concept:* 16-item scale designed to measure four dimensions of sexual self-concept: sexual self-esteem (4 items), sexual self-efficacy (4 items), assertive sexual behavior (4 items), and assertive sexual communication (4 items). Response options corresponded to behavioral/attitudinal statements in a Likert format of four ordered categories (1 = “Never” to 4 = “Always”; 1 = “Strongly disagree” to 4 = “Strongly agree”). Higher scores suggest higher levels of sexual self-concept. The scale presented evidence of validity based on the internal structure of the test and satisfactory levels of reliability in all its dimensions [46].

*Two-Dimensional Sexual Sensation Seeking Scale*: 9-item scale designed to measure sensation seeking in the sexual domain through two dimensions: sexual thrill-seeking (4 items) and tendency to sexual boredom (5 items). Response options corresponded to behavioral/attitudinal statements in a Likert format of four ordered categories (i.e., 0 = “never” to 3 = “always”). Higher scores suggest higher levels of sexual sensation seeking. The scale reported evidence of validity based on the test’s internal structure and adequate levels of reliability [47].

### 2.3. Procedure

Initially, a total of 20 surveyors were trained in the cities of Arica, Alto Hospicio, Iquique, Antofagasta, and Calama, who invited young people and young adults who passed through the busiest areas of each city to participate voluntarily, explaining the objectives of the study, and inviting them to respond on the spot. The sample collection process was carried out between March and July 2019. Those young people who chose to participate were provided with an informed consent form along with the questionnaire, which established the research objectives, confidentiality, anonymity, and the rights of the participants. Anonymity was safeguarded by returning the questionnaire in a sealed envelope without personal identification. The questionnaires were self-administered in pencil and paper format, the response procedure lasted 15–20 min, and the participants did not receive any reward in return.

The Scientific Ethics Committee of the Universidad de Tarapacá approved this research within the framework of the FONDECYT Initiation Project No. 11170395.

### 2.4. Data Analysis

Prior to the estimation of the structural equation model (SET-ESEM), the measurement models were tested and debugged through confirmatory factor analysis (CFA) and exploratory structural equation modeling (ESEM) with the WLSMV estimation method, which is robust with non-normal discrete variables [48,49]. Due to the ordinal data structure, the CFA and ESEM were also estimated from the polychoric correlation matrix [50]. Fit was assessed following the cut-point recommendations proposed by Schreiber [51] for the comparative fit index (CFI), Tucker–Lewis index (TLI), and root mean squared error of approximation (RMSEA) (e.g., CFI and TLI > 0.90 is acceptable and >0.95 is satisfactory; RMSEA < 0.08 is acceptable and <0.06 is satisfactory).

Finally, with the debugged measurement models, a SET-ESEM model of direct effects of HIV risk perception, knowledge of HIV risk behaviors, negative attitudes toward condom use, sexual self-efficacy, and sexual sensation seeking (hereafter psychological factors) on sexual risk behaviors were estimated, with covariation among the independent variables restricted only to the dimensions of each measurement model.

The SET-ESEM model was estimated from the polychoric correlation matrix using the WLSMV estimation method. The cut-off points proposed by Schreiber [51] were used for interpretation. Finally, latent variable analyses were performed with Mplus software version 8.2 [52], while descriptive analyses were performed with Jamovi software version 0.9.5.11 [53].

## 3. Results

### Measurement Models

According to the most common fit criteria in the literature (CFI > 0.95, TLI > 0.95 and RMSEA < 0.08) [51], the measurement models evidenced adequate levels of fit, and the details are presented in Table 2.

Subsequently, to identify the relationships of the explicative model, a SET-ESEM model was tested. The model presented adequate fit indexes (X^2^ = 3311.433, df = 1471, CFI = 0.964, TLI = 0.959, RMSEA = 0.036, RMSEA CI 90% = 0.035–0.038), showing it to be an adequate representation of the observed relationships.

According to the standardized effects of the psychological factors on SRB, 19 of the 33 direct effects were statistically significant, with seven mild, five moderate, and seven large effects, according to Cohen’s criteria [54]. The model explained 56% of the variance of sexual activity with multiple partners, 77% of the inadequate use of protective barriers, and 58.8% of sexual activity under the influence of alcohol or drugs. Details of the standardized model effects are presented in Table 3.

In the case of multiplicity of sexual partners, it was observed that the variables risk perception (γ = 0.197, *p* < 0.000), the affective dimension of attitudes (γ = 0.205, *p* = 0.001), sexual self-efficacy (γ = 0.444, *p* =.000), assertive sexual communication (γ = 0.116, *p* < 0.047) and sexual sensation seeking (γ = 0.578, *p* < 0.000), showed direct effects, whereas sexual self-esteem (γ = −0.308, *p* = 0.000) and assertive sexual behavior (γ = −0.162, *p* = 0.002), showed inverse effects.

Regarding the inadequate use of protective barriers, it was observed that risk perception (γ = 0.160, *p* < 0.000), the behavioral dimension of negative attitudes towards condom use (γ = 0.780, *p* < 0.000), assertive sexual communication (γ = 0.214, *p* < 0.000), and sexual sensation seeking (γ = 0.375, *p* < 0.000) presented direct effects, while knowledge about risky (γ = −0.102, *p* < 0.000) and risk-free (γ = −0.356, *p* < 0.000) behaviors, together with the cognitive dimension of negative attitudes toward condom use (γ = −0.356, *p* = 0.000) showed inverse effects.

Finally, in the case of sexual activity under the influence of alcohol and/or drugs, it was shown that the affective dimension of negative attitudes towards condom use (γ = 0.169, *p* = 0.006), sexual self-efficacy (γ = 0.295, *p* = 0.000), and sexual sensation seeking (γ = 0.666, *p* < 0.000) had direct effects, whereas the sexual self-esteem variable showed an inverse effect (γ = −0.238, *p* = 0.000).

## 4. Discussion

This study aimed to propose an explanatory model of sexual risk behaviors (i.e., inappropriate use of protective barriers, sexual activity under the influence of alcohol and drugs, and multiple sexual partners) in young people and adults due to the joint effects of psychological factors (i.e., attitude towards the use of condoms, sexual self-concept, sexual sensation seeking, knowledge of sexual risky/non-risky behaviors, and risk perception). From a statistical point of view, the proposed model seems to be a sufficient representation of the population based on the covariations observed in the sample. Therefore, it can be considered a plausible explanation for self-reported risky sexual behaviors.

In addition, the relationships showed similarities with the model based on the literature (see Figure 1), although with some exceptions. In this sense, it was observed that although knowledge is commonly indicated as a protective factor [55], the effects observed in this research were minor. In the same way, even though the perception of risk is conceptualized as one of the highly related components associated with preventive action [56], only slight effects or even the absence of evidence of population effects were shown in this research. These results are particularly interesting since the prevention programs carried out in the study country are focused on prevention information and risk perception [57], which could be insufficient in light of these results.

These results can be explained, to some extent, by interaction effects not included in this study; for example, the authors of [58] stated that the relationship between knowledge and risky sexual behaviors hinges on the perception of risk and that knowledge per se would not affect these practices.

Therefore, although both variables can be considered a necessary condition for decision making, the results in this study reflect the need to diversify the preventive actions beyond information campaigns or focused on increasing risk perceptions. These actions could include, for example, variables such as self-perception of the individuals about their sexuality, the perception of their abilities, their communication skills, and assertive sexual behavior, as well as personality changes associated with the search for sexual emotions.

However, in the results, a particular case was observed and was apparently contradictory since the cognitive dimension of attitudes presented an inverse relationship with the inadequate use of protective barriers, which is far from what is commonly found in the literature, where attitude is usually pointed out as one of the variables with the highest incidence when explaining the intention or use of condoms [23,59]. Nonetheless, after a detailed analysis of the items (i.e., “I think condoms should only be used by promiscuous people; The use of condoms is only for one-night stands”; “I think condoms are unnecessary in healthy people”; “I think that suggesting the use of condoms generates mistrust”), we believe that this discrepancy is attributable to methodological limitations of the scale used since the items could be reflecting conservative positions, rather than a general cognitive assessment towards condom use.

Finally, it is imperative to mention that this study has some limitations that must be considered. The first of these limitations is the impossibility of inferring causality, given that it was a cross-sectional study with a correlational scope; therefore, the results should be considered as an initial approximation. A second restriction is the non-probabilistic nature of the sample, which reduces the possibility of generalizing based on these findings. A third element to consider is that all the variables were measured with Likert-type, self-report, pencil, and paper scales; for this reason, the total explained variance is expected to be slightly overestimated, given the possible common fluctuation of the method. In addition, it is necessary to recognize a series of individual, economic, social, and cultural elements or factors that have not been included in the model and may have some effect on sexual risk behaviors [24].

Despite these observations, the results of this study, while providing support for most of the relationships reported in the literature between psychological factors (e.g., attitude toward condoms, sexual self-concept, sexual sensation seeking) and sexual risk behaviors, draw attention to the emphasis commonly given to some of them. Therefore, these findings emphasize the need to incorporate other psychological and behavioral variables in the study of factors that reduce sexual risk behaviors and invite the incorporation of uncommon variables (e.g., personality traits and self-assessment of their capabilities) in the different actions of health promotion and the prevention of risk behaviors in intervention strategies.

## 5. Conclusions

The results support the explanatory role of the joint effects of some psychological factors widely used in health intervention models model of risky sexual behaviors (i.e., inadequate use of protective barriers, sexual encounters under the influence of alcohol and drugs, and multiple sexual partners), in youth and young adults in northern Chile. In addition, it was shown that sexual self-concept and sexual sensation seeking are variables that notably increased the prediction power of the model, given their influence on the manifestation of sexual risk behaviors.

## Figures and Tables

**Figure 1 ijerph-19-09293-f001:**
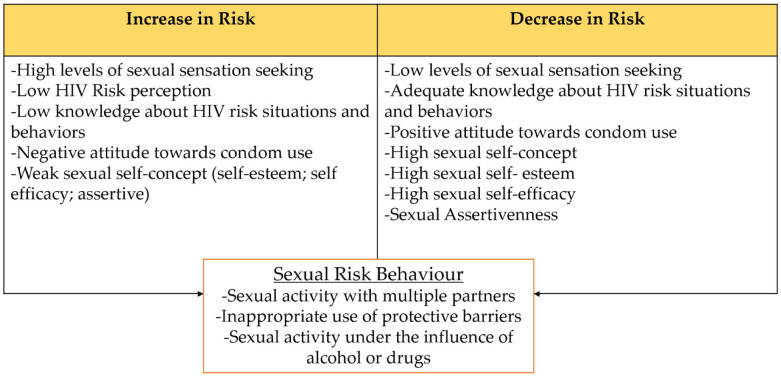
Theoretical relationships between dimensions of sexual risk behaviors and psychological factors.

**Table 1 ijerph-19-09293-t001:** Sociodemographic characteristics of the sample.

		*M* (*SD*) o *n* (%)
Biological Sex	Male	464 (47.3%)
	Female	514 (52.4%)
Age (years)		23.3 (4.68)
Marital status	Single	873 (88.0%)
	Married	74 (7.4%)
	Stable couple	28 (2.8%)
	Not reported	17 (1.8%)
Educational level	With higher education	557 (57.8%)
	No higher education	407 (42.2%)
	Not reported	28 (2.8%)
Sexual orientation	Heterosexual	818 (82.4%)
	Homosexual	28 (2.8%)
	Bisexual	34 (3.4%)
	Other	2 (0.2%)
	Not reported	110 (11.2%)
Number of reported sexual partners		6.27 (10.71)
Diagnosed with HIV/AIDS	Yes	6 (0.6%)
	No	974 (98.2%)
	Not reported	12(1,2%)
In the last 2 years, have they used condoms?	Yes, regularly	565 (56.8%)
	No	377 (37.9.0%)
	Not reported	50 (5.3%)
HIV/AIDS tests performed	Yes, regularly	447 (45.0%)
	No	527 (53.0%)
	Not reported	18 (2.0%)
HIV/AIDS tests requested to your sexual partner	Yes, regularly	304 (30.5%)
	No	659 (66.6%)
	Not reported	29 (2.9%)
Diagnosed with STI	Never	934 (94.1%)
	Only once	37 (3.7%)
	Twice	5 (0.5%)
	More than twice	16 (1.7%)
	Not reported	304 (30.5%)

**Table 2 ijerph-19-09293-t002:** Global fit of measurements models.

Model	Par	χ^2^	*df*	CFI	TLI	RMSEA	RMSEA CI 90%		SRMR
							Low	Upp	
Sexual risk behavior	51	284.544	51	0.968	0.959	0.070	0.062	0.077	0.056
HIV risk perception	16	24.506	2	0.984	0.951	0.109	0.073	0.150	0.029
Attitude towards condom use	43	234.819	32	0.975	0.964	0.082	0.072	0.092	0.042
Sexual sensation seeking	16	2.274	2	1.00	1.00	0.012	0.000	0.067	0.006
Sexual self-concept	71	944.552	98	0.971	0.964	0.096	0.090	0.101	0.040
Knowledge about HIV	44	345.345	53	0.960	0.950	0.076	0.069	0.084	0.095

Note: Par = number of parameters; χ^2^ = chi-square; *df* = degrees of freedom; *p* = significance; CFI = Comparative fit index; TLI = Tucker–Lewis index; RMSEA = Root mean square error of approximation; CI = confidence interval; Low = lower; Upp = upper; SRMR = Standarized root mean squared residual.

**Table 3 ijerph-19-09293-t003:** Standardized effects of the SET-ESEM model.

Psychological Factors	Restricted (M1)		
	SAMP	IUBP	SAIAD
HIV risk perception	0.197 **	0.160 **	0.064
Knowledge about risk behaviors	0.024	−0.102 *	−0.006
Knowledge about non-risky behaviors	0.012	−0.356 **	0.000
Negative attitudes towards the use of condoms	0.205 *	−0.097	0.169 **
Negative behaviors towards the use of condoms	0.003	0.780 **	0.130
Negative cognitions about condom use	−0.058	−0.441 **	−0.006
Sexual self-esteem	−0.308 **	−0.042	−0.238 *
Sexual self-efficacy	0.444 **	0.131	0.295 *
Assertive sexual behavior	−0.162 **	0.019	−0.093
Assertive sexual communication	0.116 **	0.214 **	0.085
Sexual sensation seeking	0.578 **	0.375 **	0.666 **

* Note: * = *p* < 0.05; ** = *p* < 0.001; SAMP = sexual activity with multiple partners; UIBP = inappropriate use of protective barriers; SAIAD = sexual activity under the influence of alcohol or drugs.

## Data Availability

Not applicable.

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
