# Peer review of "Psychological Factors and Sexual Risk Behaviors: A Multidimensional Model Based on the Chilean Population"

_ijerph, 2022, doi:10.3390/ijerph19159293_

Round 1

Reviewer 1 Report

Dear Authors,

thank you very much for the opportunity to review this interesting paper. I find the paper very useful and the issue is very relevant for the domain  of preventive choices and transformations of disadaptive and risk behaviors. I suggest to authors two aspect that, in my opnion, can improve the position of the article within the literature:

- I suggest to improve the introduction section with a clear positioning within the main paradigms, within the clinical or social health psyhcology, that can explain the issue proposed. I refer to presentation and positioning of authors among the main paradigms and theoretical frameworks that study the health risk behaviors and preventive/trasformation choices (eg. health belief model, social practive theory, theroy of access, protection motivation theory, theory of planned behaviors etc etc). 

- in addition it could interesting to know which are the main informative communication compaigns used to promote health knoledge.

- I suggest to improve the discussion and conclusion section to insert the results within the main theory of authors and propose a way to manage the social and clinical implications.

Reviewer 2 Report

1. I have revied the paper and imo, the design is approporiate, however, cultural context could be useful in the paper. For example, stating in the title that the database comes from Chile would be great - "Psychological Factors and Sexual Risk Behaviors, a multidimensional model based on Chilean population". So, what is the attitude of Chileans towards sex? Are you sex positive, or more conservative? How do you think this affects the results. Such cultural contexts are increadibly interesting in such papers and if you want to know what I'm talking about, please read the paper by Stokłosa et al. where they give cultural background in a similar study but in Poland, as it's a model one to me: https://www.mdpi.com/1660-4601/18/7/3737/htm

2. Please avoid placing the citations in the middle of the sentence, put those at the end of each one.

3. English correction is a must, please give the paper to a native English collegue or use a professional service, several typos (first graphic "INCREASE") etc.

This is a valid paper. :-)
